# Age-Related Individual Behavioural Characteristics of Adult Wistar Rats

**DOI:** 10.3390/ani11082282

**Published:** 2021-08-02

**Authors:** Sergey K. Sudakov, Elena V. Alekseeva, Galina A. Nazarova, Valentina G. Bashkatova

**Affiliations:** P.K. Anokhin Research Institute of Normal Physiology, 125315 Moscow, Russia; pravo-nadzor@yandex.ru (E.V.A.); g-a-nazarova@rambler.ru (G.A.N.); vbashkatova@inbox.ru (V.G.B.)

**Keywords:** age, behaviour, open field, physical activity, anxiety, Wistar rat

## Abstract

**Simple Summary:**

Rats are considered adults from 2 to 5 months. During this period, they are used for experimentation in physiology and pharmacology. Adult rats, depending on their age, can be in a different physiological state, which can influence the results of experiments carried out on them. Despite this, age-related changes in adult rats have not yet been examined. Our results showed that as male and female rats progressed from 2 to 5 months of age there was a decrease in the level of motor and exploratory activities, and an increase in the level of anxiety-like behaviour. Age-related changes were dependent upon initial individual characteristics of behaviour. For example, animals that demonstrated high motor activity at 2 months become significantly less active by 5 months, and animals that showed a low level of anxiety at 2 months become more anxious by 5 months. Low-activity and high-anxiety rats did not show any significant age-related changes from 2 to 5 months of age. The results of this work should be taken into account when choosing the age of rats for conducting behavioural experiments.

**Abstract:**

The aim of this work was to study age-related changes in the behaviour of adult Wistar rats using the open field (OF) and elevated plus maze (EPM) tests. Behavioural changes related to motor activity and anxiety were of particular interest. Results showed that as male and female rats progressed from 2 to 5 months of age, there was a decrease in the level of motor and exploratory activities and an increase in their level of anxiety. Age-related changes were dependent upon initial individual characteristics of behaviour. For example, animals that demonstrated high motor activity at 2 months become significantly less active by 5 months, and animals that showed a low level of anxiety at 2 months become more anxious by 5 months. Low-activity and high-anxiety rats did not show any significant age-related changes in OF and EPM tests from 2 to 5 months of age, except for a decrease in the number of rearings in the EPM. Thus, the behaviour of the same adult rat at 2 and 5 months of age is significantly different, which may lead to differences in the experimental results of physiological and pharmacological studies using adult animals of different ages.

## 1. Introduction

Approximately 80% of the experimental non genetically modified animals used for physiological and pharmacological studies are mice, rats, and guinea pigs [1,2]. Rats are the animals most frequently used in scientific research [3]. Moreover, rats are frequently used to study the mechanisms underlying the functioning of the nervous system, as well as in behavioural studies [4].

Currently, one of the most widely used lines of laboratory animals is the Wistar rat. A significant number of modern laboratory rat lines have been developed from this line, including Sprague Dawley and Long Evans rats. It is known that various behavioural characteristics of rats change with age. The average lifespan of a rat is about 3 years [5]. Rats develop rapidly and their adolescence ends by the end of the second month of ontogenesis. Thus, a rat at 2 months of age (60 days) is considered an adult [2,5,6,7]. By this time, all the vital systems of the rat’s body have matured. The social maturity of rats occurs between the ages of 5 and 6 months [2,5]. After 6–7 months, rats begin to show signs of ageing. As each individual grows and develops, its behaviour also changes.

In rodents, age-related changes in behaviour are usually investigated by comparing the corresponding indicators of young and old animals. Compared to younger rats, a number of studies have reported that older rats have a decrease in behavioural characteristics associated with sensory, motor, and cognitive functions [8,9,10]. When studying the behaviour of rats in the elevated plus maze (EPM) test, it was found that older rats showed a higher level of anxiety [11]. Testing rats of different ages in the Morris water maze showed a decrease in cognitive function in elderly compared to young rats [12]. It was previously shown that adolescent rats had a higher level of anxiety in the EPM and open field (OF) tests compared to adult animals [13]. However, although most physiological and pharmacological studies are conducted in adult rats (2 to 5 months), there has not yet been a specific study of the behavioural changes that occur during this period of a rat’s lifespan.

It has been shown that rats, although identical in lineage (breed), sex, age, and conditions of housing, can individually differ significantly in terms of behaviour and physiology [14,15]. Such individual differences can be quite stable and can be detected using behavioural screening tests [16,17,18,19]. It has been shown that initial individual differences can influence the formation of age-related changes in behaviour [20,21,22]. However, these data are relevant to old, but not younger (i.e., aged 2–5 months) adult rats.

The aim of this work was to study age-related changes in the behaviour of adult Wistar rats aged 2 to 5 months old using the EPM and OF tests. Age-related changes occurring in animals with individual characteristics at the level of motor activity and anxiety were also studied.

## 2. Materials and Methods

### 2.1. Animals

The experiments were carried out on 50 male and 27 female Wistar rats weighing 210–240 g (male) and 170–200 g (female), which were obtained from Stolbovaya Animal Clinic (Moscow, Russia). The animals were housed in ventilated cages (Techniplast (Milan, Italy) green line 1500U) with natural corn bedding (Zolotoy Kot, Voronezh, Russia), 4–5 individuals in each cage, with free access to water and a standard combined food (3 kcal/g; Profgryzun, Moscow, Russia) at a temperature of 21 °C in the presence of lighting (90 Lux) from 20:00 to 08:00.

### 2.2. Experimental Tests

At the age of 2 months (61–62 days), 30 males and 27 females (group 1) were studied in two successive experimental tests: OF and EPM. Tests were carried out at intervals of 24 h.

The OF was a circular arena 98 cm in diameter, surrounded by walls 31 cm high (OpenScience, Moscow, Russia, model TS0501-R). The arena was divided by lines into 7 central and 12 peripheral sectors (Figure 1). The arena was uniformly illuminated by a medical lamp (illumination of the arena surface = 450 Lux). For the OF test, a rat was placed in the centre of the arena and behaviour was observed for five minutes, and the following endpoints were recorded: horizontal activity (total number of crossed squares); vertical activity (rearings), and time spent in the centre of the arena. The animal was returned to its home cage after the test.

The EPM test (The Elevated Plus Maze, Columbus Instruments, Chicago, IL USA) uses a cruciform platform with four arms (length = 50 cm, width = 15 cm). The two opposite arms of the maze had high, opaque walls, whereas the other two were open. The height of the walls of the closed arms was 43 cm. The maze was raised to a height of 75 cm. The central part of the EPM was a 15 × 15 cm square. Illumination of the surface of the maze was 90 Lux. For this test, a rat was placed for 5 min in the centre of the EPM platform and the total locomotor activity was determined by the total number of crossings episodes from one arm of the EPM to another, the number of rearings, as well as the individual anxiety of the animals by the time spent on the open arms of the EPM [23]. All experiments were carried out between 09:00 and 18:00.

The remaining 20 males (group 2) at the age of 2 months were not studied in these tests. Upon reaching the age of 5 months (153–154 days), animals from the first group were again investigated by OF and EPM tests, whereas animals from the second group were studied in these tests at this age for the first time.

### 2.3. Division into Subgroups

Subgroups were created from rats of group 1 at the ages of 2 months by selecting 7–8 with the highest and lowest values for: motor activity in the OF (subgroups 1.1.high-m, 1.1.low-m, 1.1.high-f, and 1.1.low-f) and EPM (subgroups 1.2.high-m, 1.2.low-m, 1.2.high-f, and 1.2.low-f) tests; time spent in the centre of the OF (subgroups 1.3.HA-m, 1.3.LA-m, 1.3.HA-f, and 1.3.LA-f), and time spent on the open arms of the EPM (subgroups 1.4.HA-m, 1.4.LA-m, 1.4.HA-f, and 1.4.LA-f). All recorded parameters were compared separately for all males and females, as well as in separate groups of males with high and low motor activity or anxiety at the ages of 2 and 5 months.

### 2.4. Statistics

Statistics were performed using the statistical software Statistica 10.0. The data obtained in each experimental subgroup were assessed for normality using the Shapiro–Wilk and Kolmogorov–Smirnov tests. Since the test showed the absence of a normal distribution, the calculations were carried out by the nonparametric Mann–Whitney U test. Results in tables and figures are presented as the mean ± SD. The results were assessed as significant at *p* < 0.05.

### 2.5. Ethical Statement

The protocols and procedures for this study were ethically reviewed and approved by the Animal Care and Use Committee of the P.K. Anokhin Research Institute of Normal Physiology (Permission number 336) and conform to Directive 2010/63/EU.

## 3. Results

### 3.1. Age-Related Changes in Adult Rats

It was found that in 5-month-old males who performed behavioural tests for the first time (first group) in comparison with 2-month-old males, there was a significant decrease in motor activity as measured by the EPM (U = 116.50; Z = 3.29; *p* = 0.00995) and OF (U = 74; Z = 4.20; *p* = 0.0003) tests. In 5-month-old males who were tested a second time (second group), there was also a decrease in motor activity (U = 155; Z = 3.61; *p* = 0.0003) and a decrease in the number of rearings (U = 92.50; Z = 4.69; *p* = 0.000003) in the EPM, and a reduction in motor activity (U = 63.50; Z = 5.20; *p* = 0.001) and number of rearings (U = 203.50; Z = 2.77; *p* = 0.005) in OF (Figure 2A,B).

In females, by the fifth month, a significant decrease in motor activity in the OF test (U = 46; Z = 3.65; *p* = 0.0002) and a decrease in the number of rearings in the EPM (U = 36.50; Z = 3.65; *p* = 0.00008) were observed (Figure 3A,B).

Since there was little variation in OF and EPM results at the age of 5 months in males in the first group compared to males in the second group at the same age, further analysis of age-related changes in the same animals was carried out during the initial testing at the age of 2 months and retesting at the age of 5 months. In females, comparison of single and repeated testing was not carried out.

### 3.2. Age-Related Changes in Rats with High and Low Motor Activity in the OF and EPM Tests

For males, locomotor activity decreased with age, as determined by the OF (U = 0.01; Z = 2.93; *p* = 0.003) and EPM (U = 4; Z = 2.55; *p* = 0.01) tests. No significant differences were found in the level of anxiety. For males that were passive in the OF test, no significant age-related changes were observed except for a decrease in the number of rearings in the EPM (U = 5.0 Z = 2,427,731 *p* = 0.015194) (Table 1).

In active females, motor activity decreased with age in the OF test (U = 0.01; Z = 3.06; *p* = 0.002). However, these changes were less pronounced than in males. In females that were passive in the OF, only a decrease in the number of rearings in the EPM was noted (U = 7.50 Z = 2,108,293 *p* = 0.035006) (Table 2).

The active and passive males in the OF test at the age of 2 months differed only in the level of horizontal (U = 0.01; Z = −3.07; *p* = 0.02) and vertical activity (U = 0.01; Z = −3.06; *p* = 0.002) (Table 1). At the age of 5 months, these differences diminished. In active and passive females at the age of 5 months, the differences observed at 2 months in the level of motor activity in the OF (U = 0.01; Z = −3.06; *p* = 0.002) also receded (Table 2).

In males that were active in the EPM, a significant decrease with age in motor activity was also observed, both in the OF (U = 0.01; Z = 3.06; *p* = 0.002) and in the EPM (U = 0.01; Z = 3.06; *p* = 0.002). The number of rearings also decreased in the OF (U = 8; Z = 2.04; *p* = 0.04) and in the EPM (U = 1.0; Z = 2.9; *p* = 0.003) tests. There were no age-related differences in the level of anxiety.

In males that were passive in the EPM, no significant age-related changes in the levels of motor and exploratory activities were observed (Table 3).

In females active in the EPM, motor activity significantly decreased with age in the OF (U = 7.50; Z = 2.11; *p* = 0.03) and EPM (U = 1; Z = 2.94; *p* = 0.003) tests, and the number of rearings in the EPM decreased (U = 2; Z = 2.81; *p* = 0.005). No significant age-related changes were observed in passive females (Table 4).

Males, both active and passive in the EPM at the age of 2 months significantly differed in the number of intersections of the maze compartments (U = 0.01; Z = −3.07; *p* = 0.002). At 5 months, these differences were not apparent.

EPM motor activity in females at 2 months was significantly greater compared to passive females (U = 0.01; Z = −2.93; *p* = 0.003). They also spent longer time in the open arms (U = 1.50; Z = −2.71; *p* = 0.007) than rats that were passive in the EPM. At 5 months, these differences between active and passive females in the EPM were not apparent. 

### 3.3. Age-Related Changes in Rats with High (HA) and Low Levels of Anxiety (LA) in the OF and EPM Tests

Despite the fact that we did not reveal age-related differences in the level of anxiety in the general population of Wistar rats (Table 1 and Table 2), LA in the OF test males, at the age of 2 months, demonstrated a low level of anxiety (spent a longer time in the centre of the OF), and had a significantly increased level of anxiety by the fifth month (U = 11.50; Z = 2.10; *p* = 0.036). In addition, in LA males in the OF, by the fifth month, motor activity decreased both in the OF (U = 0.01; Z = 3.31; *p* = 0.0009) and in the EPM (U = 7; Z = 2.57; *p* = 0.01) tests, and the number of rearings in the OF (U = 8.50; Z = 2.41; *p* = 0.02) and in the EPM (U = 8; Z = 2.47; *p* = 0.01) also decreased (Table 5). Those males that demonstrated a high level of anxiety at age 2 months had no reliable age-related changes in levels of motor activity and anxiety. We observed only a significant decrease in the number of rearings in the EPM (U = 3.0; Z = 2.68; *p* = 0.007).

In both LA and HA in the OF test females, no significant age-related changes in the studied parameters were observed, except for a decrease in the number of rearings in the EPM in calm (U = 3.5; Z =2.62; *p* =0.009) and in anxious animals (U =7.5; Z = 2.11; *p* = 0.035) (Table 6).

The observed significant differences between LA and HA in the OF test in males at the age of 2 months in motor activity (U = 1; Z = −3.07; *p* = 0.002), the number of rearings (U =5.5; Z = −2.55; *p* = 0.01), and the time spent in the centre of the arena (U = 0.01; Z = −3.18; *p* = 0.001) was not present by the fifth month. The observed significant differences between LA and HA 2-month-old females in terms of the time spent in the centre of the OF (U = 0.01; Z = −3.07; *p* = 0.002) was also not apparent at the age of 5 months.

Males who, at the age of 2 months, demonstrated a low level of anxiety in the EPM (prolonged time spent in open arms), by 5 months significantly reduced the time spent on open arms (U = 7.0; Z = 2.17; *p* = 0.03). Motor and exploratory activity of LA in EPM males, both in the EPM (U = 7.5; Z = 2.11; *p* =0.035) and in the OF (U = 1; Z = 2.94; *p* = 0.003) also significantly decreased with age. In males that demonstrated a high level of anxiety at the age of 2 months, the number of rearings in the EPM decreased with age (U = 2.5; Z = 2.75; *p* = 0.006). No other significant age-related changes in behaviour were observed (Table 7). In males that demonstrated a high level of anxiety in the EPM at the age of 2 months, practically no significant age-related changes in behaviour were observed.

Females showing a low level of anxiety in the EPM at the age of 2 months also significantly reduced the time spent on open arms by 5 months (U = 4.0; Z = 2.555; *p* = 0.01). In addition, with age, they had a decrease in the number of rearings in the EPM (U = 2.5; Z = 2.75; *p* = 0.006). In females showing a high level of anxiety in the EPM, by the fifth month the time spent in the open arms did not change. However, the number of rearings in the EPM significantly decreased (U = 3.0; Z = 2.68; *p* = 0.007) (Table 8).

Males with high and low anxiety in the EPM differed at the age of 2 months in the duration of stay in the open arms of the maze (U = 0.01; Z = −3.07; *p* = 0.002). By the fifth month, significant differences between groups were not observed.

Females with high and low anxiety in the EPM at 2 months significantly differed in the time spent on the open arms of the maze (U = 0.0 1; Z = −3.07; *p* = 0.002), as well as in the level of motor activity in the EPM (U = 3; Z = −2.68; *p* = 0.007). These differences were not found at the fifth month.

## 4. Discussion

The obtained results showed that in adulthood, both in male and female Wistar rats, there were significant decreases in motor and exploratory activities. Arrant et al. (2013) found a decrease in activity and an increase in the level of anxiety in rats from 1 to 2 months of age [24], and Boguszewski and Zagrodzka (2002) described a decrease in motor activity and an increase in anxiety in rats from 4 to 24 months [25]. Thus, we can postulate that the decrease in activity (in active rats) and increase in anxiety (in LA rats) begins at least in late adolescence and continues into adulthood and old age.

The results of our studies show that age-related changes in the behaviour of adult rats depend on their initial individual characteristics. Thus, the greatest age-related changes were observed in animals with high motor activity and low levels of anxiety. On the contrary, in rats with reduced activity and a high level of anxiety, practically no changes were observed. These findings have not been previously described.

If we assume that by the age of 2 months the main neurochemical systems that support motor activity and facilitate anxiety are formed, then we can surmise that the decrease in activity and increase in anxiety by the fifth month is associated with social influences occurring during this interval. Indeed, it has been shown that social isolation can increase the level of anxiety in animals [26,27,28] and in humans [29] and can have a significant effect on the level of motor activity. Thus, social isolation led to hyperactivity in males [30,31,32] and females [33]. In contrast, an enriched social environment reduces the level of anxiety in animals [34]. Increasing evidence suggests that effects of chronic stress on behavioural performance show that sex of the subject, as well as duration and intensity of stress, is an important determinant of the behavioural, neurochemical, and anatomical consequences of the stress [35]. Furthermore, stress differentially affected central transmitter levels in the frontal cortex, hippocampus, and amygdala depending on age [36] and sex [37]. In our experiments, animals were neither socially isolated nor kept in an enriched environment. The rats were kept in standard, ventilated cages, at 4–5 animals per cage. Nevertheless, even this level of social spacing led to changes in the behaviour of the animals. Apparently, the establishment and maintenance of hierarchical relationships can also have a stressful effect on animals [38].

The results of the second study with OF and EPM tests in males at the age of 5 months did not differ very much from the initial results (Figure 2). This suggests that the animals did not recall the previous testing, and therefore the changes observed in the behaviour of animals in these tests can be considered due to age, and not repetition of the procedure. The only significant difference was a reduction in the number of rearings during repeated testing, compared to a single test, both in the OF and EPM tests. It is likely that upon repeated testing, an additional decrease in exploratory activity occurs due to an insignificant retention of the memory of the animal’s stay in these experimental conditions at the age of 2 months. However, it is improbable that these differences can have a significant effect on the occurrence of age-related changes in the behaviour of rats, since there were no significant differences between these parameters during single and repeated testing.

Results showed that females had approximately the same age-related changes in adulthood as males. The individual characteristics of the behaviour of female Wistar rats also changed in a manner similar to that of males with ageing. However, we only performed a repeated study of females in OF and EPM tests. A possible confounding factor for females could be that they are in different estrous cycles. On the one hand, Scholl et al. (2019) showed the absence of any influence of the estrous cycle on the behaviour of females in OF and EPM tests [39]. On the other hand, it has been shown that ovarian hormones have a significant effect on the behaviour of rats [40]. Females in diestrus have been shown to exhibit anxiety-like behaviours while females in metaestrus behaved in a similar way as males [41]. We did not investigate what state of the estrous cycle the animals were in during the experiment. In this regard, we cannot consider the age-related changes in the behaviour of females obtained in our experiment as absolutely reliable. However, the similarity with the results obtained with males allows us to make a cautious conclusion about the absence of sex differences in the occurrence of age-dependent changes in the behaviour of adult Wistar rats.

Thus, the behaviour of the same adult rat at 2 and 5 months of age is significantly different, which may lead to differences in the experimental results of physiological and pharmacological studies using adult animals of different age.

## 5. Conclusions

In male and female Wistar rats there is a decrease in the level of motor and exploratory activities from 2 to 5 months of age.Age-related changes in adult Wistar rats depend on their initial individual characteristics of behaviour.Animals that demonstrate high motor activity at 2 months become significantly less active by 5 months.Animals that show a low level of anxiety at the age of 2 months become more anxious by 5 months.Low-activity and high-anxiety rats do not exhibit age-related changes in OF and EPM tests from 2 to 5 months of age, except for a decrease in the number of rearings in the EPM.

## Figures and Tables

**Figure 1 animals-11-02282-f001:**
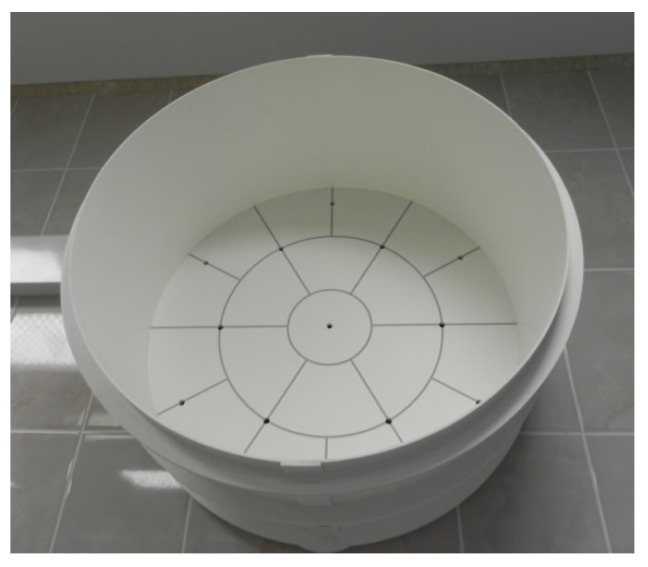
The OF arena.

**Figure 2 animals-11-02282-f002:**
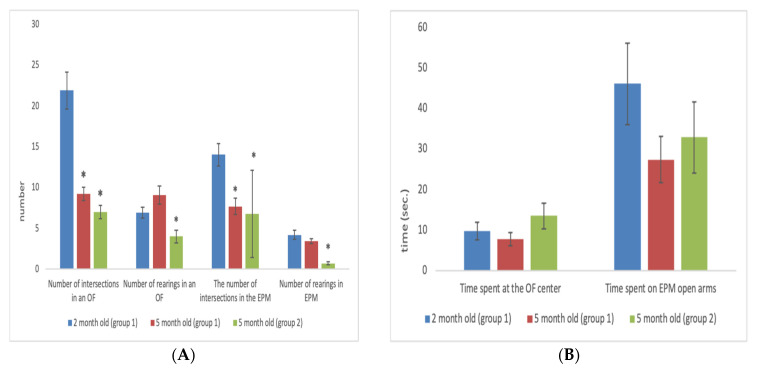
(**A**,**B**). Behavioural activity of males at the ages of 2 and 5 months. *****
*p* < 0.05 (2-month-old vs. 5-month-old). (**A**): Values in numbers. (**B**): Values in seconds.

**Figure 3 animals-11-02282-f003:**
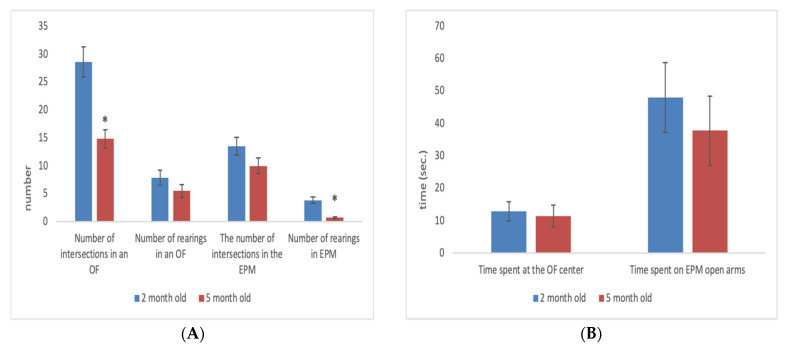
(**A**,**B**). Behavioural activity of females at the ages of 2 and 5 months. *****
*p* < 0.05 (2-month-old vs. 5-month-old). (**A**): Values in numbers. (**B**): Values in seconds.

**Table 1 animals-11-02282-t001:** Behavioural activity of males, active and passive in the OF test (subgroups 1.1).

-	Number of Intersections in an OF	Number of Rearings in an OF	Time Spent at the OF Centre	The Number of Intersections in the EPM	Time Spent on EPM Open Arms	Number of Rearings in EPM
2-month-old (subgroup 1.1.high-m)	37 ± 3.70	9.29 ± 0.96	16.6 ± 3.7	13.43 ± 1.5	47.43 ± 17.2	4.9 ± 0.94
5-month-old (subgroup 1.1.high-m)	9.1 ± 1.43 *	4.6 ± 1.58	11.7 ± 3.75	6.29 ± 1.30 *	24.9 ± 13.46	0.9 ± 0.31 *
2-month-old (subgroup 1.1.low-m)	9.15 ± 1.48 ^#^	3.29 ± 0.6 ^#^	2.9 ± 1.33 ^#^	10 ± 3.05	35 ± 11.9	3.1 ± 0.47
5-month-old (subgroup 1.1.low-m)	5.15 ± 1.27	1.58 ± 0.72	13.9 ± 5.5	5.29 ± 1.64	28.29 ± 19,6	0.6 ± 0.53 *

******p* < 0.05 (2-month-old vs. 5-month-old); ^#^
*p* < 0.05 (subgroup 1.1.high-m vs. subgroup 1.1.low-m).

**Table 2 animals-11-02282-t002:** Behavioural activity of females, active and passive in the OF test (subgroups 1.1).

-	Number of Intersections in an OF	Number of Rearings in an OF	Time Spent at the OF Centre	The Number of Intersections in the EPM	Time Spent on EPM Open Arms	Number of Rearings in EPM
2-month-old (subgroup 1.1.high-f)	40.86 ± 2.28	11 ± 2.58	18.15 ± 5.68	16 ± 3.02	56.29 ± 15.36	3.29 ± 0.48
5-month-old (subgroup 1.1.high-f)	15.15 ± 3.13 *	6.58 ± 1.95	16 ± 6.42	11.29 ± 1.57	56.34 ± 16.26	0.58 ± 0.28 *
2-month-old (subgroup 1.1.low-f)	17.72 ± 0.94 ^#^	5.15 ± 1.32	10.72 ± 3.74	11.29 ± 1.72	50.43 ± 21.69	4 ± 1.21
5-month-old (subgroup 1.1.low-m-f)	14 ± 2.75	5.58 ± 1.85	8.58 ± 4.37	10 ± 2.71	38.43 ± 19.17	0.58 ± 0.28 *

******p* < 0.05 (2-month-old vs. 5-month-old); ^#^
*p* < 0.05 (subgroup 1.1.high-f vs. subgroup 1.1.low-f).

**Table 3 animals-11-02282-t003:** Behavioural activity of males, active and passive in the EPM test (subgroups 1.2).

-	Number of Intersections in an OF	Number of Rearings in an OF	Time Spent at the OF Centre	The Number of Intersections in the EPM	Time Spent on EPM Open Arms	Number of Rearings in EPM
2-month-old (subgroup 1.2.high-m)	20.86 ± 2.29	8.29 ± 1.45	15.29 ± 7.22	23 ± 0.89	57 ± 10.79	4.15 ± 0.43
5-month-old (subgroup 1.2.high-m)	6.72 ± 1.49 *	3.43 ± 1.41 *	7.43 ± 4.64	6.86 ± 2.13 *	38.72 ± 19.37	0.15 ± 0.13 *
2-month-old (subgroup 1.2.low-m)	14.58 ± 4.31	4.85 ± 1.28	4.72 ± 2.22	4.71 ± 1.04 ^#^	46 ± 31.09	2.58 ± 0.60
5-month-old (subgroup 1.2.low-m)	7.58 ± 1.47	3.43 ± 1.24	24.29 ± 9.10	7.43 ± 1.92	45.86 ± 21.64	1 ± 0.53

******p* < 0.05 (2-month-old vs. 5-month-old); ^#^
*p* < 0.05 (subgroup 1.2.high-m vs. subgroup 1.2.low-m).

**Table 4 animals-11-02282-t004:** Behavioural activity of females that were active and passive in the EPM test (subgroups 1.2).

-	Number of Intersections in an OF	Number of Rearings in an OF	Time Spent at the OF Centre	The Number of Intersections in the EPM	Time Spent on EPM Open Arms	Number of Rearings in EPM
2-month-old (subgroup 1.2.high-f)	31.86 ± 3.68	11.43 ± 2.11	10.86 ± 2.60	20.58 ± 1.55	69.86 ± 13.67	4.72 ± 0.80
5-month-old (subgroup 1.2.high-f)	16.72 ± 2.24 *	7.86 ± 1.57	8.43 ± 2.79	11.58 ± 1.79 *	50.58 ± 18.23	0.86 ± 0.43 *
2-month-old (subgroup 1.2.low-f)	24.84 ± 4.26	4 ± 1.13 ^#^	17.17 ± 7.46	6.5 ± 0.66 ^#^	20 ± 5.7#	2 ± 0.75
5-month-old (subgroup 1.2.low-f)	11.17 ± 3.48	2.17 ± 1.32 ^#^	18.67 ± 7.93	8.17 ± 2.50	34 ± 19.66	0.5 ± 0.20

******p* < 0.05 (2-month-old vs. 5-month-old); ^#^
*p* < 0.05 (subgroup 1.2.high-f vs. subgroup 1.2.low-f).

**Table 5 animals-11-02282-t005:** Behavioural activity of LA and HA in the OF test males, (subgroups 1.3).

-	Number of Intersections in an OF	Number of Rearings in an OF	Time Spent at the OF Centre	The Number of Intersections in the EPM	Time Spent on EPM Open Arms	Number of Rearings in EPM
2-month-old (subgroup 1.3. LA-m)	33.88 ± 4.67	10.25 ± 0.65	22.63 ± 5.15	14 ± 1.70	40.38 ± 17.05	4 ± 1.03
5-month-old (subgroup 1.3. LA-m)	8.25 ± 1.49 *	4.25 ± 1.11 *	13 ± 3.92 *	6.88 ± 1.38 *	37.63 ± 12.78	0.62 ± 0.28 *
2-month-old (subgroup 1.3. HA-m)	9.72 ± 1.82 ^#^	4.15 ± 1.19 ^#^	0.57 ± 0.34 ^#^	11.15 ± 3.43	31 ± 11.18	4.72 ± 0.92
5-month-old (subgroup 1.3. HA-m)	5.86 ± 1.33	2.29 ± 0.94	4.43 ± 3.85	5.72 ± 2.05	25.29 ± 19.95	0.72 ± 0.52 *

******p* < 0.05 (2-month-old vs. 5-month-old); ^#^
*p* < 0.05 (subgroup 1.2. LA-m vs. subgroup 1.2. HA-m).

**Table 6 animals-11-02282-t006:** Behavioural activity of LA and HA in the OF test females (subgroups 1.3).

-	Number of Intersections in an OF	Number of Rearings in an OF	Time Spent at the OF Centre	The Number of Intersections in the EPM	Time Spent on EPM Open Arms	Number of Rearings in EPM
2-month-old (subgroup 1.3. LA-f)	24.29 ± 4.63	8.29 ± 1.80	24.29 ± 4.63	9.72 ± 1.87	41.86 ± 22.47	3.29 ± 0.85
5-month-old (subgroup 1.3. LA-f)	19.2 ± 2.11	5.86 ± 1.77	17 ± 8.11	9.58 ± 2.70	43 ± 18.61	0.43 ± 0.28 *
2-month-old (subgroup 1.3. HA-f)	3 ± 0.97	4 ± 1.12	3 ± 0.97 ^#^	12.43 ± 1.74	37.43 ± 7.48	4.14 ± 1.15
5-month-old (subgroup 1.3. HA-f)	8.72 ± 1 *^#^	2.15 ± 0.74	6.29 ± 2.56	8.43 ± 1.99	34.86 ± 19.44	0.58 ± 0.19 *

******p* < 0.05 (2-month-old vs. 5-month-old); ^#^
*p* < 0.05 (subgroup 1.3. LA-f vs. subgroup 1.3. HA-f).

**Table 7 animals-11-02282-t007:** Behavioural activity of LA and HA in the EPM test males (subgroups 1.4).

-	Number of Intersections in an OF	Number of Rearings in an OF	Time Spent at the OF Centre	The Number of Intersections in the EPM	Time Spent on EPM Open Arms	Number of Rearings in EPM
2-month-old (subgroup 1.4. LA-m)	20.4 ± 4.38	5.7 ± 1.26	6.3 ± 2.09	17.4 ± 3.12	111.9 ± 23.46	3.4 ± 0.67
5-month-old (subgroup 1.4. LA-m)	5.4 ± 1.24 *	1.4 ± 0.49 *	16 ± 8.61	5.7 ± 1.53 ± *	38.1 ± 20.35 *	0.3 ± 0.17 *
2-month-old (subgroup 1.4. HA-m)	16.4 ± 3.96	6.3 ± 1.10	5.7 ± 2.34	8.1 ± 1.77	2 ± 0.93 ^#^	5 ± 0.61
5-month-old (subgroup 1.4. HA-m)	8.4 ± 1.29	5.6 ± 1.74	18 ± 5.48	9 ± 1.59	18.1 ± 8.92	1.3 ± 0.52 *

******p* < 0.05 (2-month-old vs. 5-month-old); # *p* < 0.05 (subgroup 1.4. LA-m vs. subgroup 1.4. HA-fm).

**Table 8 animals-11-02282-t008:** Behavioural activity of LA and HA in the EPM test females (subgroups 1.4).

-	Number of Intersections in an OF	Number of Rearings in an OF	Time Spent at the OF Centre	The Number of Intersections in the EPM	Time Spent on EPM Open Arms	Number of Rearings in EPM
2-month-old (subgroup 1.4. LA-f)	29.72 ± 3.78	8.72 ± 2.64	11.43 ± 2.65	19.15 ± 2.17	90.43 ± 17.37	3.86 ± 0.47
5-month-old (subgroup 1.4. LA-f)	16.86 ± 1.98	7.86 ± 1.57	8.15 ± 2.90	12.43 ± 2.33	27.86 ± 7.90 *	1 ± 0.40*
2-month-old (subgroup 1.4. HA-f)	28.43 ± 5.99	7.58 ± 3.39	19.15 ± 5.99	8.58 ± 1.26#	11 ± 2.17 ^#^	4 ± 0.99
5-month-old (subgroup 1.4.HA-f)	11 ± 3.26 *	3.72 ± 1.21	14.86 ± 7.22	7.72 ± 2.13	13.33 ± 8.05	0.43 ± 0.19 *

******p* < 0.05 (2-month-old vs. 5-month-old); ^#^
*p* < 0.05 (subgroup 1.4. LA-f vs. subgroup 1.4. HA-f).

## Data Availability

The data presented in this study are available on request from the corresponding author. The data are not publicly available due to the rules of the P.K. Anokhin Research Institute of Normal Physiology.

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
