# Peer review of "Age-Related Individual Behavioural Characteristics of Adult Wistar Rats"

_animals, 2021, doi:10.3390/ani11082282_

Round 1

Reviewer 1 Report

This manuscript compares male and female Wistar rats in terms of possible age-related behavioural differeces.

Overall, I do agree that rats of different age and sex may potentially present different behavioural features. Of course such an aspect is able to seriously impact behavioral studies. That said, various serious issues, in this study, do need a very careful revision before it can be published.

The first, and probably most important problem in this study is of methodological order and it concerns the organization of groups and sub-groups. Another important concern I have is that  results are very difficult to be appreciated even for an experienced reader. A detailed list of comments and suggestions is below reported.

INTRODUCTION

Overall clear and sufficiently well written. 

>> "It has been shown that rats, although identical [...] and physiology". - This is an extremely important statement underlying the rationale of this study. Please add suitable references here.

METHODS

The organization of groups and subgroups is difficult to follow.

>> Groups and sub-groups are not homogeneous. It should be better discussed why do, overall, 50 males and 27 females have been used? 

>> Similarly, it should be better discussed why do the first group encompass 30 males and 27 females and the second group 20 subjects.

>> What does it mean that "Subgroups were created by selecting 7-8 rats with the highest and lowest values for [...]"?  - Does it mean that OF and EPM tests have been performed and that, on the basis of results obtained, subgroups organized? Such an aspect is misunderstanding and must be clarified.

>> It is unclear how "active" and "passive" subjects have been defined. This is a critical aspect and must be carefully explained and clarified

>> It is unclear how "calm" and "anxious" subjects have been defined. This is a critical aspect and must be carefully explained and clarified

>> The terms "passive" and "calm" are uncommon and misunderstanding. What does it mean that a rat is... "calm"?

RESULTS

>> Actually the study presents its results only by means of an exceedingly high number tables. In brief,  it is very difficult to see differences between subjects/groups even for a experienced reader. It is suggested the utilization of histograms to reduce the utilization of tables. Tables are important but, in this case, it is also important to presents results in a more intelligible/comprehensible fashion. 

DISCUSSION

The discussion is exceedingly brief and concise. If on the one hand, sometimes, a concise and brief discussion may represent a positive aspect, on the other hand, in the current form, various aspects must be better developed in this manuscript

>> Paragraph Lines 239-250 "If we assume [...] stressful effect on animals [32]" - This is an important aspect. Please develop and, importantly, add more suitable references

>> Paragraph Lines 262-276 " Result showed that [...] adult Wistar rats" - This paragraph is difficult to understand. In addition contrasting findings are also presented (refs 33, 34, 35). 

Author Response

Response to Reviewer 1 Comments

I would like to express my gratitude to the reviewer for constructive criticism, which made it possible to make the article much better. Below are the answers to comments and remarks:

>> "It has been shown that rats, although identical [...] and physiology". - This is an extremely important statement underlying the rationale of this study. Please add suitable references here.

Done.

METHODS

The organization of groups and subgroups is difficult to follow.

We tried to explain in more detail the creating the subgroups in Materials and Methods and name the groups with letters.

>> Groups and sub-groups are not homogeneous. It should be better discussed why do, overall, 50 males and 27 females have been used? 

>> Similarly, it should be better discussed why do the first group encompass 30 males and 27 females and the second group 20 subjects.

We used two groups of males, 30 (first group) and 20 (second group). There are more animals in the first group, because we have created subgroups from this group. No subgroups were  formed from the second subgroup. There is only one group of females - 27 rats. Subgroups were created from it, as well as from the first group of males. Initially, 30 female rats were obtaned, the same number as males. However, 3 females were excluded from the experiment, due to the presence of dyspeptic symptoms during quarantine. In our opinion, this did not lead to a distortion of the results obtained, since we used the nonparametric Mann-Whitney U test, which makes it possible to use non-homogeneous groups 

>> What does it mean that "Subgroups were created by selecting 7-8 rats with the highest and lowest values for [...]"?  - Does it mean that OF and EPM tests have been performed and that, on the basis of results obtained, subgroups organized? Such an aspect is misunderstanding and must be clarified.

We tried to explain in more detail the creating the subgroups in Materials and Methods

>> It is unclear how "active" and "passive" subjects have been defined. This is a critical aspect and must be carefully explained and clarified

Active subjects were 7-8 rats at the age of 2 months, selected from Group 1 with the highest number of sector intersections in the OF (subgroups 1.1.high-m and 1.1.high-f ) or in EPM (subgroups 1.2.high-m and 1.2.high-f) . Passive, on the contrary, had the smallest value of intersections in the OF (subgroups 1.1.low-m and 1.1.low-f ) or in the EPM (subgroups 1.2.low-m and 1.2.low-f)

>> It is unclear how "calm" and "anxious" subjects have been defined. This is a critical aspect and must be carefully explained and clarified

>> The terms "passive" and "calm" are uncommon and misunderstanding. What does it mean that a rat is... "calm"?

We have corrected the terms calm and anxious to animals with low (LA) and high levels of anxiety (HA). Subjects  with low level of anxiety were 7-8 rats at the age of 2 months, selected from Group 1 with the highest time spent at the OF center (subgroups 1.3.LA-m and 1.3.LA-f ) or in open arms of EPM (subgroups 1.3.LA-m and 1.3.LA-f) . Rats with high level of anxiety had the smallest value of time spent at the OF center (subgroups 1.3.HA-m and 1.3.HA-f ) or time spent on EPM open arms (subgroups 1.4.HA-m and 1.4.HA-f). 

RESULTS

>> Actually the study presents its results only by means of an exceedingly high number tables. In brief,  it is very difficult to see differences between subjects/groups even for a experienced reader. It is suggested the utilization of histograms to reduce the utilization of tables. Tables are important but, in this case, it is also important to presents results in a more intelligible/comprehensible fashion. 

Tables 1 and 2 are replaced by figures. The rest of the tables, in our opinion, should be left as they contain main information.

DISCUSSION

The discussion is exceedingly brief and concise. If on the one hand, sometimes, a concise and brief discussion may represent a positive aspect, on the other hand, in the current form, various aspects must be better developed in this manuscript

>> Paragraph Lines 239-250 "If we assume [...] stressful effect on animals [32]" - This is an important aspect. Please develop and, importantly, add more suitable references

Done

>> Paragraph Lines 262-276 " Result showed that [...] adult Wistar rats" - This paragraph is difficult to understand. In addition contrasting findings are also presented (refs 33, 34, 35). 

This paragraph contains information that the results of our experiments on females may not be completely reliable. Therefore, we consider the main results of experiments carried out on males, despite the fact that the data obtained on females are similar to those of males.

Reviewer 2 Report

The authors of the manuscript Age-related individual behavioral characteristics of adult Wistar rats address one major important point in the focus of nowadays research: the topic of individuality of experimental animals. Two age-classes of Wistar rats were tested, with 2 and 5 months to address young and older (but not very old) adult rats. So the focus is on the age when most rats are used for (behavioral) studies, when animals are considered to reflect a stable ‘normal’ state in terms of behavior and hormonal status. 

Although the topic is of interest, the study has severe flaws which have to be addressed.

Introduction:

  1. 1st paragraph: rats are commonly not the animals most used, as mice and zebrafish are used more often because of their genetically modified status. Please correct or specify (e.g. most used genetically not modified animals etc.).
  2. As your main focus and all discussion is based on the fact that rats are mature with 2 months, please specify with references this date. As ‘month’ is a not well-defined timespan (28-31 days in calendars or 4 weeks with 7 days each) and therefore varying between 61 days (30+31) and 58 days (8 weeks with 7 days), this has to be well defined. As Klein and Romeo (2013, Hormones and Behavior) specify, adolescence is the phase in rats between 50-60 days, a time definition of 2 months=58 days would mean rats of late adolescence were tested in this study. This would completely change the topic of the study (not meaning making it less important but rather different).
  3. Please check references 3 and 4-6 as titles do not seem to fit to the cited context.

Materials and Methods:

Please follow the ARRIVE guidelines to describe your animals, housing and experimental procedures. Many questions are then answered:

  1. How can you keep 27 females in groups per 5 animals?
  2. Housing conditions are missing completely (cage, bedding, food, water, light intensity, temperature, humidity and much more). Please check the ARRIVE guidelines (also available as checklist) to address everything.
  3. I do not understand the sectors of your openfield. Perhaps a sketch or photo would be helpful, as well as from the EPM.
  4. Your light conditions are quite high and extreme to test activity in albino rats, as they are very light sensitive. Please comment why this was chosen.
  5. It would be easier to understand the different groups of rats (males) when this is more clear who was tested when and how often. The description is ok, but perhaps you name the groups more stringent or with e.g. letters, be creative!
  6. Please define ‘number of movements’ along the arms. Is this how often they reached the end?
  7. Your EPM with open arms with 3cm walls is rather unusual, please cite references ising this as well.
  8. Subgroups: Am I right to understand that subgroup 1 actually consists of 2 groups namely performing high or low?
  9. Subgroups: can animals be selected for more than 1 subgroup?
  10. Can animals switch the group by age?
  11. Extremely important and highly discussed in the research community is the topic of experimental units. As you do not “treat” animals indicudualy with any applications, your experimental unit would be the cage and not the individual animal. This would, from my point of view, make your point rather difficult to state. But on this problem you have to make a statement why exp. units are not possible to take into account or at least try to do so. Perhaps you find that cage means are even more stable and hide the effect, or support the effect, because variance gets down. This is very important to consider and discuss.
  12. Stats: The complete study would benefit from (paired) correlations as they take into account this huge amount of animals without selecting only few for subgroup comparisons.

Results:

  1. Subheadings: as you have only behavioral results, you can remove ‘in behavioral parameters’. I would recommend to use ‘general’ age-related in heading 3.1.
  2. You define ‘activity’ but it is not really clear for me which accounts to this: vertical/horizontal movement? Please specify.
  3. All tables miss explaining text telling the units, if the +/- is SD or SEM and what * or # means. Perhaps this was lost. Please add.
  4. Please specify ‘variation’.- SD, SEM, variance? Why not calculating the co-efficient of variation (easy to use and to understand) when discussing variation? As high variance is often discussed as critical point in aging animals (and females tow because of different hormonal status), this would be a nice add-on.
  5. Unfortunately, for me rigor/logic of low variation to the idea of forming subgroups to subdivide the groups with low variation in groups with even lower variation (and much lower n-numbers) is not clear for me. Please explain.
  6. 2: I understand that these results are for comparisons within the subgroups and not between, but this needs some time. Perhaps you try to make this easier to get.
  7. Please define what ‘calm’ is. I would recommend not to use this as it sounds very anthropocentric for me (calm = introverted) but first it is the absent of noise. It would be fine for me if you stick to this term, when it is the most perfect fitting for you, but please define well which parameters were used to define a rat to join the calm sub-group, especially when you compare to ‘anxious’.

Discussion:

  1. Line 235: initial individual characteristics: this assumes clearly now for me that subgroups are determined by young adult behavior. (would be the answer to my topic no. 10, but rather late).
  2. You surmise about social influence (hierarchy, social behavior in general) which is well written, but what about other physiological states, like hormones, decreasing neurotransmitters, or effects of impoverished housing conditions (no enrichment).
  3. The comparision of the second study OF/EPM in males was not clear to me to find in the results section. Please indicate the heading so it is easier to find the table/section.
  4. Line 259 ff: all experiences in lifetime are forming the individual. It is known that already single experiences can impact the following generations via epigenetic effects. As I understand that your rats are tested once in their lifetime in an openfield and once in an EPM and then kept impoverished for 3 months without any changes/challenges in their daily life, I would rather assume that this single experience would have a great impact.
  5. Conclusions: as for my understanding, variance is normally increasing with age, I would add this exception from your study, as it shows that this is perhaps also an effect of very low animal numbers.

Author Response

Response to Reviewer 2 Comments

I would like to thank the reviewer for constructive comments and suggestions, which allowed not only to improve this article, but also to create plans for the future. Below are the answers to comments:

Introduction:

  1. 1st paragraph: rats are commonly not the animals most used, as mice and zebrafish are used more often because of their genetically modified status. Please correct or specify (e.g. most used genetically not modified animals etc.).

Done

  1. As your main focus and all discussion is based on the fact that rats are mature with 2 months, please specify with references this date. As ‘month’ is a not well-defined timespan (28-31 days in calendars or 4 weeks with 7 days each) and therefore varying between 61 days (30+31) and 58 days (8 weeks with 7 days), this has to be well defined. As Klein and Romeo (2013, Hormones and Behavior) specify, adolescence is the phase in rats between 50-60 days, a time definition of 2 months=58 days would mean rats of late adolescence were tested in this study. This would completely change the topic of the study (not meaning making it less important but rather different).

Done

  1. Please check references 3 and 4-6 as titles do not seem to fit to the cited context.

References were replaced.

Materials and Methods:

Please follow the ARRIVE guidelines to describe your animals, housing and experimental procedures. Many questions are then answered:

  1. How can you keep 27 females in groups per 5 animals?

Corrected

  1. Housing conditions are missing completely (cage, bedding, food, water, light intensity, temperature, humidity and much more). Please check the ARRIVE guidelines (also available as checklist) to address everything.

Corrected

  1. I do not understand the sectors of your openfield. Perhaps a sketch or photo would be helpful, as well as from the EPM.

Photo included

  1. Your light conditions are quite high and extreme to test activity in albino rats, as they are very light sensitive. Please comment why this was chosen.

It was mistake in description of procedure. In fact, illumination was 450 Lux.

  1. It would be easier to understand the different groups of rats (males) when this is more clear who was tested when and how often. The description is ok, but perhaps you name the groups more stringent or with e.g. letters, be creative!

Done

  1. Please define ‘number of movements’ along the arms. Is this how often they reached the end?

Corrected

  1. Your EPM with open arms with 3cm walls is rather unusual, please cite references ising this as well.

It was mistake, corrected

  1. Subgroups: Am I right to understand that subgroup 1 actually consists of 2 groups namely performing high or low?

Corrected. Each subgroup has separate name

  1. Subgroups: can animals be selected for more than 1 subgroup?

Yes, some animals were selected for more than 1 group. However, in the compared groups, the same animals were not present.

  1. Can animals switch the group by age?

No. Rats were in the same group from 2 to 5 months.

  1. Extremely important and highly discussed in the research community is the topic of experimental units. As you do not “treat” animals indicudualy with any applications, your experimental unit would be the cage and not the individual animal. This would, from my point of view, make your point rather difficult to state. But on this problem you have to make a statement why exp. units are not possible to take into account or at least try to do so. Perhaps you find that cage means are even more stable and hide the effect, or support the effect, because variance gets down. This is very important to consider and discuss.

In this article we only compare the selected groups of animals with same individual characteristics of behavior. This allowed us to carry out statistics and determine which individual characteristics of rats are most susceptible to changes with age. We use term “individual” because we selected animals into the groups according their individual characteristics, obtained in OF and EPM individually (not by cage). 

  1. Stats: The complete study would benefit from (paired) correlations as they take into account this huge amount of animals without selecting only few for subgroup comparisons.

There were no any significant correlations between most of behavioral activity in rats of group 1 at the ages of 2 and 5 months.  Except is the time spent on EPM open arms, where the correlation is over 0.9. (see Table). This interesting fact requires a separate study and additional experiments. We descried do not include this data to this article, but discuss it in separate article, when the picture will be more clear.

Spearman's rank correlation coefficient

Number of intersections in an OF

Number of rearings in an OF

Time spent at the OF center

The number of intersections in the EPM

Time spent on EPM open arms

Number of rearings in EPM

Male

0,316474

0,369312

-0,0518

0,976265

0,073149

0,258896

Female

0,028838

0,325386

0,647474

0,971544

0,023149

0,166718

Results:

  1. Subheadings: as you have only behavioral results, you can remove ‘in behavioral parameters’. I would recommend to use ‘general’ age-related in heading 3.1.

Done

  1. You define ‘activity’ but it is not really clear for me which accounts to this: vertical/horizontal movement? Please specify.

Done

  1. All tables miss explaining text telling the units, if the +/- is SD or SEM and what * or # means. Perhaps this was lost. Please add.

Done

  1. Please specify ‘variation’.- SD, SEM, variance? Why not calculating the co-efficient of variation (easy to use and to understand) when discussing variation? As high variance is often discussed as critical point in aging animals (and females tow because of different hormonal status), this would be a nice add-on.

The subject of this article was to compare subgroups with the same behavioral characteristics, so we removed everything related to the term "variation".

  1. Unfortunately, for me rigor/logic of low variation to the idea of forming subgroups to subdivide the groups with low variation in groups with even lower variation (and much lower n-numbers) is not clear for me. Please explain.

We formed subgroups according to one of the behavioral indicators (for example, the time spent on the open arms of the EPM). We selected 7-8 animals with the highest and lowest values from 30 males. Thus, two subgroups were formed (1.4. LA-m and 1.4. HA-m).

  1. Please define what ‘calm’ is. I would recommend not to use this as it sounds very anthropocentric for me (calm = introverted) but first it is the absent of noise. It would be fine for me if you stick to this term, when it is the most perfect fitting for you, but please define well which parameters were used to define a rat to join the calm sub-group, especially when you compare to ‘anxious’.

Done

Discussion:

  1. Line 235: initial individual characteristics: this assumes clearly now for me that subgroups are determined by young adult behavior. (would be the answer to my topic no. 10, but rather late).
  2. You surmise about social influence (hierarchy, social behavior in general) which is well written, but what about other physiological states, like hormones, decreasing neurotransmitters, or effects of impoverished housing conditions (no enrichment).

Done

  1. The comparision of the second study OF/EPM in males was not clear to me to find in the results section. Please indicate the heading so it is easier to find the table/section.

Done

  1. Line 259 ff: all experiences in lifetime are forming the individual. It is known that already single experiences can impact the following generations via epigenetic effects. As I understand that your rats are tested once in their lifetime in an openfield and once in an EPM and then kept impoverished for 3 months without any changes/challenges in their daily life, I would rather assume that this single experience would have a great impact.

We assumed it was possible. Therefore, we studied the behavior of rats of the second group (Figure 1), which were not exposed to any influences at 2 months. We did not find any significant differences. This is discussed on the lines 321-331.

  1. Conclusions: as for my understanding, variance is normally increasing with age, I would add this exception from your study, as it shows that this is perhaps also an effect of very low animal numbers.

You're absolutely right, the variation increases with age. The table shows the coefficients of variation for 2 and 5 month old males. The subject of this article was to compare subgroups with the same behavioral characteristics, so we removed everything related to the term "variation". The coefficients of variation, as well as the correlation analysis, will be given in the future publication after additional experiments. 

The coefficient of variation (male)

Number of intersections in an OF

Number of rearings in an OF

Time spent at the OF center

The number of intersections in the EPM

Time spent on EPM open arms

Number of rearings in EPM

 2 month old (group 1)

0,11

10%

0,10

10%

0,23

23%

0,10

10%

0,22

22%

0,13

13%

5 month old (group 1)

0,12

12%

0,20

20%

0,32

32%

0,14

13%

0,27

27%

0,27

27%

 The coefficient of variation (female)

Number of intersections in an OF

Number of rearings in an OF

Time spent at the OF center

The number of intersections in the EPM

Time spent on EPM open arms

Number of rearings in EPM

 2 month old

0,10

9%

0,18

17%

0,23

22%

0,12

12%

0,23

22%

0,15

15%

Round 2

Reviewer 1 Report

From my reading it is clear that the authors did their best to improve their manuscript. I have no further comments.